# Deep Learning for Precipitation Nowcasting: A Benchmark and A New Model

**Xingjian Shi, Zhihan Gao, Leonard Lausen, Hao Wang, Dit-Yan Yeung**
Department of Computer Science and Engineering
Hong Kong University of Science and Technology
{xshiab,zgaoag,lelausen,hwangaz,dyyeung}@cse.ust.hk

**Wai-kin Wong, Wang-chun Woo**
Hong Kong Observatory
Hong Kong, China
{wkwong,wcwoo}@hko.gov.hk

## Abstract

With the goal of making high-resolution forecasts of regional rainfall, precipitation nowcasting has become an important and fundamental technology underlying various public services ranging from rainstorm warnings to flight safety. Recently, the *Convolutional LSTM* (ConvLSTM) model has been shown to outperform traditional optical flow based methods for precipitation nowcasting, suggesting that deep learning models have a huge potential for solving the problem. However, the convolutional recurrence structure in ConvLSTM-based models is *location-invariant* while natural motion and transformation (e.g., rotation) are *location-variant* in general. Furthermore, since deep-learning-based precipitation nowcasting is a newly emerging area, clear evaluation protocols have not yet been established. To address these problems, we propose both a new model and a benchmark for precipitation nowcasting. Specifically, we go beyond ConvLSTM and propose the *Trajectory GRU* (TrajGRU) model that can actively learn the *location-variant* structure for recurrent connections. Besides, we provide a benchmark that includes a real-world large-scale dataset from the Hong Kong Observatory, a new training loss, and a comprehensive evaluation protocol to facilitate future research and gauge the state of the art.

## 1 Introduction

Precipitation nowcasting refers to the problem of providing very short range (e.g., 0-6 hours) forecast of the rainfall intensity in a local region based on radar echo maps[1], rain gauge and other observation data as well as the *Numerical Weather Prediction* (NWP) models. It significantly impacts the daily lives of many and plays a vital role in many real-world applications. Among other possibilities, it helps to facilitate drivers by predicting road conditions, enhances flight safety by providing weather guidance for regional aviation, and avoids casualties by issuing citywide rainfall alerts. In addition to the inherent complexities of the atmosphere and relevant dynamical processes, the ever-growing need for real-time, large-scale, and fine-grained precipitation nowcasting poses extra challenges to the meteorological community and has aroused research interest in the machine learning community [23, 25].

The conventional approaches to precipitation nowcasting used by existing operational systems rely on optical flow [28]. In a modern day nowcasting system, the convective cloud movements are first estimated from the observed radar echo maps by optical flow and are then used to predict the future radar echo maps using semi-Lagrangian advection. However, these methods are unsupervised from the machine learning point of view in that they do not take advantage of the vast amount of existing radar echo data. Recently, progress has been made by utilizing supervised *deep learning* [15] techniques for precipitation nowcasting. Shi *et al.* [23] formulated precipitation nowcasting as a spatiotemporal sequence forecasting problem and proposed the *Convolutional Long Short-Term Memory* (ConvLSTM) model, which extends the LSTM [7] by having convolutional structures in both the input-to-state and state-to-state transitions, to solve the problem. Using the radar echo sequences for model training, the authors showed that ConvLSTM is better at capturing the spatiotemporal correlations than the fully-connected LSTM and gives more accurate predictions than the *Real-time Optical flow by Variational methods for Echoes of Radar* (ROVER) algorithm [28] currently used by the Hong Kong Observatory (HKO).

However, despite their pioneering effort in this interesting direction, the paper has some deficiencies. First, the deep learning model is only evaluated on a relatively small dataset containing 97 rainy days and only the nowcasting skill score at the 0.5mm/h rain-rate threshold is compared. As real-world precipitation nowcasting systems need to pay additional attention to heavier rainfall events such as rainstorms which cause more threat to the society, the performance at the 0.5mm/h threshold (indicating raining or not) alone is not sufficient for demonstrating the algorithm's overall performance [28]. In fact, as the area *Deep Learning for Precipitation Nowcasting* is still in its early stage, it is not clear how models should be evaluated to meet the need of real-world applications. Second, although the convolutional recurrence structure used in ConvLSTM is better than the fully-connected recurrent structure in capturing spatiotemporal correlations, it is not optimal and leaves room for improvement. For motion patterns like rotation and scaling, the local correlation structure of consecutive frames will be different for different spatial locations and timestamps. It is thus inefficient to use convolution which uses a *location-invariant* filter to represent such *location-variant* relationship. Previous attempts have tried to solve the problem by revising the output of a recurrent neural network (RNN) from the raw prediction to be some location-variant transformation of the input, like optical flow or dynamic local filter [5, 3]. However, not much research has been conducted to address the problem by revising the recurrent structure itself.

In this paper, we aim to address these two problems by proposing both a benchmark and a new model for precipitation nowcasting. For the new benchmark, we build the HKO-7 dataset which contains radar echo data from 2009 to 2015 near Hong Kong. Since the radar echo maps arrive in a stream in the real-world scenario, the nowcasting algorithms can adopt online learning to adapt to the newly emerging patterns dynamically. To take into account this setting, we use two testing protocols in our benchmark: the offline setting in which the algorithm can only use a fixed window of the previous radar echo maps and the online setting in which the algorithm is free to use all the historical data and any online learning algorithm. Another issue for the precipitation nowcasting task is that the proportions of rainfall events at different rain-rate thresholds are highly imbalanced. Heavier rainfall occurs less often but has a higher real-world impact. We thus propose the *Balanced Mean Squared Error* (B-MSE) and *Balanced Mean Absolute Error* (B-MAE) measures for training and evaluation, which assign more weights to heavier rainfalls in the calculation of MSE and MAE. We empirically find that the balanced variants of the loss functions are more consistent with the overall nowcasting performance at multiple rain-rate thresholds than the original loss functions. Moreover, our experiments show that training with the balanced loss functions is essential for deep learning models to achieve good performance at higher rain-rate thresholds. For the new model, we propose the *Trajectory Gated Recurrent Unit* (TrajGRU) model which uses a subnetwork to output the state-to-state connection structures before state transitions. TrajGRU allows the state to be aggregated along some learned trajectories and thus is more flexible than the *Convolutional GRU* (ConvGRU) [2] whose connection structure is fixed. We show that TrajGRU outperforms ConvGRU, *Dynamic Filter Network* (DFN) [3] as well as 2D and 3D *Convolutional Neural Networks* (CNNs) [20, 27] in both a synthetic MovingMNIST++ dataset and the HKO-7 dataset.

Using the new dataset, testing protocols, training loss and model, we provide extensive empirical evaluation of seven models, including a simple baseline model which always predicts the last frame, two optical flow based models (ROVER and its nonlinear variant), and four representative deep learning models (TrajGRU, ConvGRU, 2D CNN, and 3D CNN). We also provide a large-scale

benchmark for precipitation nowcasting. Our experimental validation shows that (1) all the deep learning models outperform the optical flow based models, (2) TrajGRU attains the best overall performance among all the deep learning models, and (3) after applying online fine-tuning, the models tested in the online setting consistently outperform those in the offline setting. To the best of our knowledge, this is the first comprehensive benchmark of deep learning models for the precipitation nowcasting problem. Besides, since precipitation nowcasting can be viewed as a video prediction problem [22, 27], our work is *the first* to provide evidence and justification that online learning could potentially be helpful for video prediction in general.

## 2 Related Work

**Deep learning for precipitation nowcasting and video prediction** For the precipitation nowcasting problem, the reflectivity factors in radar echo maps are first transformed to grayscale images before being fed into the prediction algorithm [23]. Thus, precipitation nowcasting can be viewed as a type of video prediction problem with a fixed "camera", which is the weather radar. Therefore, methods proposed for predicting future frames in natural videos are also applicable to precipitation nowcasting and are related to our paper. There are three types of general architecture for video prediction: RNN based models, 2D CNN based models, and 3D CNN based models. Ranzato *et al.* [22] proposed the first RNN based model for video prediction, which uses a convolutional RNN with $1 \times 1$ state-state kernel to encode the observed frames. Srivastava *et al.* [24] proposed the LSTM encoder-decoder network which uses one LSTM to encode the input frames and another LSTM to predict multiple frames ahead. The model was generalized in [23] by replacing the fully-connected LSTM with ConvLSTM to capture the spatiotemporal correlations better. Later, Finn *et al.* [5] and De Brabandere *et al.* [3] extended the model in [23] by making the network predict the transformation of the input frame instead of directly predicting the raw pixels. Ruben *et al.* [26] proposed to use both an RNN that captures the motion and a CNN that captures the content to generate the prediction. Along with RNN based models, 2D and 3D CNN based models were proposed in [20] and [27] respectively. Mathieu *et al.* [20] treated the frame sequence as multiple channels and applied 2D CNN to generate the prediction while [27] treated them as the depth and applied 3D CNN. Both papers show that *Generative Adversarial Network* (GAN) [6] is helpful for generating sharp predictions.

**Structured recurrent connection for spatiotemporal modeling** From a higher-level perspective, precipitation nowcasting and video prediction are intrinsically spatiotemporal sequence forecasting problems in which both the input and output are spatiotemporal sequences [23]. Recently, there is a trend of replacing the fully-connected structure in the recurrent connections of RNN with other topologies to enhance the network's ability to model the spatiotemporal relationship. Other than the ConvLSTM which replaces the full-connection with convolution and is designed for dense videos, the *SocialLSTM* [1] and the *Structural-RNN* (S-RNN) [11] have been proposed sharing a similar notion. SocialLSTM defines the topology based on the distance between different people and is designed for human trajectory prediction while S-RNN defines the structure based on the given spatiotemporal graph. All these models are different from our TrajGRU in that our model actively *learns* the recurrent connection structure. Liang *et al.* [17] have proposed the *Structure-evolving LSTM*, which also has the ability to learn the connection structure of RNNs. However, their model is designed for the semantic object parsing task and learns how to merge the graph nodes automatically. It is thus different from TrajGRU which aims at learning the local correlation structure for spatiotemporal data.

**Benchmark for video tasks** There exist benchmarks for several video tasks like online object tracking [29] and video object segmentation [21]. However, there is no benchmark for the precipitation nowcasting problem, which is also a video task but has its unique properties since radar echo map is a completely different type of data and the data is highly imbalanced (as mentioned in Section 1). The large-scale benchmark created as part of this work could help fill the gap.

## 3 Model

In this section, we present our new model for precipitation nowcasting. We first introduce the general encoding-forecasting structure used in this paper. Then we review the ConvGRU model and present our new TrajGRU model.

## 3.1 Encoding-forecasting Structure

We adopt a similar formulation of the precipitation nowcasting problem as in [23]. Assume that the radar echo maps form a spatiotemporal sequence $\langle \mathcal{I}_1, \mathcal{I}_2, \ldots \rangle$. At a given timestamp $t$, our model generates the most likely $K$-step predictions, $\hat{\mathcal{I}}_{t+1}, \hat{\mathcal{I}}_{t+2}, \ldots, \hat{\mathcal{I}}_{t+K}$, based on the previous $J$ observations including the current one: $\mathcal{I}_{t-J+1}, \mathcal{I}_{t-J+2}, \ldots, \mathcal{I}_t$. Our encoding-forecasting network first encodes the observations into $n$ layers of RNN states: $\mathcal{H}_t^1, \mathcal{H}_t^2, \ldots, \mathcal{H}_t^n = h(\mathcal{I}_{t-J+1}, \mathcal{I}_{t-J+2}, \ldots, \mathcal{I}_t)$, and then uses another $n$ layers of RNNs to generate the predictions based on these encoded states: $\hat{\mathcal{I}}_{t+1}, \hat{\mathcal{I}}_{t+2}, \ldots, \hat{\mathcal{I}}_{t+K} = g(\mathcal{H}_t^1, \mathcal{H}_t^2, \ldots, \mathcal{H}_t^n)$. Figure 1 illustrates our encoding-forecasting structure for $n = 3, J = 2, K = 2$. We insert downsampling and upsampling layers between the RNNs, which are implemented by convolution and deconvolution with stride. The reason to reverse the order of the forecasting network is that the high-level states, which have captured the global spatiotemporal representation, could guide the update of the low-level states. Moreover, the low-level states could further influence the prediction. This structure is more reasonable than the previous structure [23] which does not reverse the link of the forecasting network because we are free to plug in additional RNN layers on top and no skip-connection is required to aggregate the low-level information. One can choose any type of RNNs like ConvGRU or our newly proposed TrajGRU in this general encoding-forecasting structure as long as their states correspond to tensors.

## 3.2 Convolutional GRU

The main formulas of the ConvGRU used in this paper are given as follows:

$$
\begin{aligned}
\mathcal{Z}_t &= \sigma(\mathcal{W}_{xz} * \mathcal{X}_t + \mathcal{W}_{hz} * \mathcal{H}_{t-1}), \\
\mathcal{R}_t &= \sigma(\mathcal{W}_{xr} * \mathcal{X}_t + \mathcal{W}_{hr} * \mathcal{H}_{t-1}), \\
\mathcal{H}'_t &= f(\mathcal{W}_{xh} * \mathcal{X}_t + \mathcal{R}_t \circ (\mathcal{W}_{hh} * \mathcal{H}_{t-1})), \\
\mathcal{H}_t &= (1 - \mathcal{Z}_t) \circ \mathcal{H}'_t + \mathcal{Z}_t \circ \mathcal{H}_{t-1}.
\end{aligned}
\tag{1}
$$

The bias terms are omitted for notational simplicity. '$*$' is the convolution operation and '$\circ$' is the Hadamard product. Here, $\mathcal{H}_t, \mathcal{R}_t, \mathcal{Z}_t, \mathcal{H}'_t \in \mathbb{R}^{C_h \times H \times W}$ are the memory state, reset gate, update gate, and new information, respectively. $\mathcal{X}_t \in \mathbb{R}^{C_i \times H \times W}$ is the input and $f$ is the activation, which is chosen to be leaky ReLU with negative slope equals to 0.2 [18] througout the paper. $H, W$ are the height and width of the state and input tensors and $C_h, C_i$ are the channel sizes of the state and input tensors, respectively. Every time a new input arrives, the reset gate will control whether to clear the previous state and the update gate will control how much the new information will be written to the state.

## 3.3 Trajectory GRU

When used for capturing spatiotemporal correlations, the deficiency of ConvGRU and other ConvRNNs is that the connection structure and weights are fixed for all the locations. The convolution operation basically applies a *location-invariant* filter to the input. If the inputs are all zero and the reset gates are all one, we could rewrite the computation process of the new information at a specific location $(i, j)$ at timestamp $t$, i.e, $\mathcal{H}'_{t,:,i,j}$, as follows:

$$
\mathcal{H}'_{t,:,i,j} = f(\mathbf{W}_{hh} \text{concat}(\langle \mathcal{H}_{t-1,:,p,q} \mid (p, q) \in \mathcal{N}_{i,j}^h \rangle)) = f\left( \sum_{l=1}^{|\mathcal{N}_{i,j}^h|} \mathbf{W}_{hh}^l \mathcal{H}_{t-1,:,p_{l,i,j},q_{l,i,j}} \right). \tag{2}
$$

Here, $\mathcal{N}_{i,j}^h$ is the ordered neighborhood set at location $(i, j)$ defined by the hyperparameters of the state-to-state convolution such as kernel size, dilation and padding [30]. $(p_{l,i,j}, q_{l,i,j})$ is the $l$th neighborhood location of position $(i, j)$. The concat$(\cdot)$ function concatenates the inner vectors in the set and $\mathbf{W}_{hh}$ is the matrix representation of the state-to-state convolution weights.

As the hyperparameter of convolution is fixed, the neighborhood set $\mathcal{N}_{i,j}^h$ stays the same for all locations. However, most motion patterns have different neighborhood sets for different locations. For example, rotation and scaling generate flow fields with different angles pointing to different directions. It would thus be more reasonable to have a location-variant connection structure as

$$
\mathcal{H}'_{t,:,i,j} = f\left( \sum_{l=1}^{L} \mathbf{W}_{hh}^l \mathcal{H}_{t-1,:,p_{l,i,j}(\theta),q_{l,i,j}(\theta)} \right), \tag{3}
$$

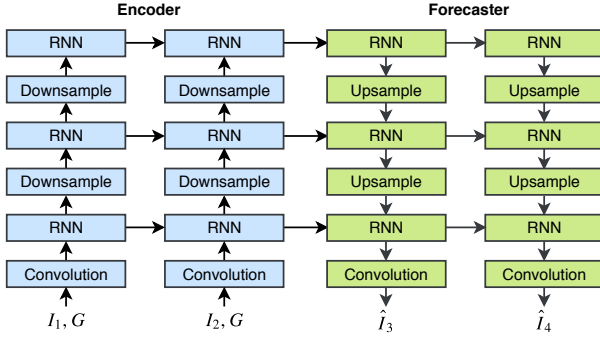

**Figure 1:** Example of the encoding-forecasting structure used in the paper. In the figure, we use three RNNs to predict two future frames $\hat{I}_3, \hat{I}_4$ given the two input frames $I_1, I_2$. The spatial coordinates $G$ are concatenated to the input frame to ensure the network knows the observations are from different locations. The RNNs can be either ConvGRU or TrajGRU. Zeros are fed as input to the RNN if the input link is missing.

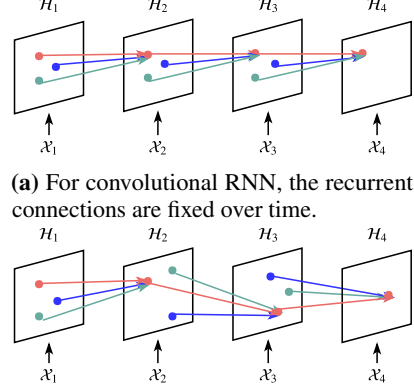

**(a)** For convolutional RNN, the recurrent connections are fixed over time.

**(b)** For trajectory RNN, the recurrent connections are dynamically determined.

**Figure 2:** Comparison of the connection structures of convolutional RNN and trajectory RNN. Links with the same color share the same transition weights. (Best viewed in color)

where $L$ is the total number of local links, $(p_{l,i,j}(\theta), q_{l,i,j}(\theta))$ is the $l$th neighborhood parameterized by $\theta$.

Based on this observation, we propose the TrajGRU, which uses the current input and previous state to generate the local neighborhood set for each location at each timestamp. Since the location indices are discrete and non-differentiable, we use a set of continuous optical flows to represent these "indices". The main formulas of TrajGRU are given as follows:

$$\mathcal{U}_t, \mathcal{V}_t = \gamma(\mathcal{X}_t, \mathcal{H}_{t-1}),$$

$$\mathcal{Z}_t = \sigma(\mathcal{W}_{xz} * \mathcal{X}_t + \sum_{l=1}^{L} \mathcal{W}_{hz}^l * \mathrm{warp}(\mathcal{H}_{t-1}, \mathcal{U}_{t,l}, \mathcal{V}_{t,l})),$$

$$\mathcal{R}_t = \sigma(\mathcal{W}_{xr} * \mathcal{X}_t + \sum_{l=1}^{L} \mathcal{W}_{hr}^l * \mathrm{warp}(\mathcal{H}_{t-1}, \mathcal{U}_{t,l}, \mathcal{V}_{t,l})), \qquad (4)$$

$$\mathcal{H}_t' = f(\mathcal{W}_{xh} * \mathcal{X}_t + \mathcal{R}_t \circ (\sum_{l=1}^{L} \mathcal{W}_{hh}^l * \mathrm{warp}(\mathcal{H}_{t-1}, \mathcal{U}_{t,l}, \mathcal{V}_{t,l}))),$$

$$\mathcal{H}_t = (1 - \mathcal{Z}_t) \circ \mathcal{H}_t' + \mathcal{Z}_t \circ \mathcal{H}_{t-1}.$$

Here, $L$ is the total number of allowed links. $\mathcal{U}_t, \mathcal{V}_t \in \mathbb{R}^{L \times H \times W}$ are the flow fields that store the local connection structure generated by the structure generating network $\gamma$. The $\mathcal{W}_{hz}^l, \mathcal{W}_{hr}^l, \mathcal{W}_{hh}^l$ are the weights for projecting the channels, which are implemented by $1 \times 1$ convolutions. The $\mathrm{warp}(\mathcal{H}_{t-1}, \mathcal{U}_{t,l}, \mathcal{V}_{t,l})$ function selects the positions pointed out by $\mathcal{U}_{t,l}, \mathcal{V}_{t,l}$ from $\mathcal{H}_{t-1}$ via the bilinear sampling kernel [10, 9]. If we denote $\mathcal{M} = \mathrm{warp}(\mathcal{I}, \mathbf{U}, \mathbf{V})$ where $\mathcal{M}, \mathcal{I} \in \mathbb{R}^{C \times H \times W}$ and $\mathbf{U}, \mathbf{V} \in \mathbb{R}^{H \times W}$, we have:

$$\mathcal{M}_{c,i,j} = \sum_{m=1}^{H} \sum_{n=1}^{W} \mathcal{I}_{c,m,n} \max(0, 1 - |i + \mathbf{V}_{i,j} - m|) \max(0, 1 - |j + \mathbf{U}_{i,j} - n|). \qquad (5)$$

The advantage of such a structure is that we could learn the connection topology by learning the parameters of the subnetwork $\gamma$. In our experiments, $\gamma$ takes the concatenation of $\mathcal{X}_t$ and $\mathcal{H}_{t-1}$ as the input and is fixed to be a one-hidden-layer convolutional neural network with $5 \times 5$ kernel size and 32 feature maps. Thus, $\gamma$ has only a small number of parameters and adds nearly no cost to the overall computation. Compared to a ConvGRU with $K \times K$ state-to-state convolution, TrajGRU is able to learn a more efficient connection structure with $L < K^2$. For ConvGRU and TrajGRU, the number of model parameters is dominated by the size of the state-to-state weights, which is $O(L \times C_h^2)$ for TrajGRU and $O(K^2 \times C_h^2)$ for ConvGRU. If $L$ is chosen to be smaller than $K^2$, the

**Table 1:** Comparison of TrajGRU and the baseline models in the MovingMNIST++ dataset. 'Conv-K$\alpha$-D$\beta$' refers to the ConvGRU with kernel size $\alpha \times \alpha$ and dilation $\beta \times \beta$. 'Traj-L$\lambda$' refers to the TrajGRU with $\lambda$ links. We replace the output layer of the ConvGRU-K5-D1 model to get the DFN.

| | Conv-K3-D2 | Conv-K5-D1 | Conv-K7-D1 | Traj-L5 | Traj-L9 | Traj-L13 | TrajGRU-L17 | DFN | Conv2D | Conv3D |
|---|---|---|---|---|---|---|---|---|---|---|
| #Parameters | 2.84M | 4.77M | 8.01M | 2.60M | 3.42M | 4.00M | 4.77M | 4.83M | 29.06M | 32.52M |
| Test MSE $\times 10^{-2}$ | 1.495 | 1.310 | 1.254 | 1.351 | 1.247 | 1.170 | **1.138** | 1.461 | 1.681 | 1.637 |
| Standard Deviation $\times 10^{-2}$ | 0.003 | 0.004 | 0.006 | 0.020 | 0.015 | 0.022 | 0.019 | 0.002 | 0.001 | 0.002 |

number of parameters of TrajGRU can also be smaller than the ConvGRU and the TrajGRU model is able to use the parameters more efficiently. Illustration of the recurrent connection structures of ConvGRU and TrajGRU is given in Figure 2. Recently, Jeon & Kim [12] has used similar ideas to extend the convolution operations in CNN. However, their proposed *Active Convolution Unit* (ACU) focuses on the images where the need for location-variant filters is limited. Our TrajGRU focuses on videos where location-variant filters are crucial for handling motion patterns like rotations. Moreover, we are revising the structure of the recurrent connection and have tested different number of links while [12] fixes the link number to 9.

## 4 Experiments on MovingMNIST++

Before evaluating our model on the more challenging precipitation nowcasting task, we first compare TrajGRU with ConvGRU, DFN and 2D/3D CNNs on a synthetic video prediction dataset to justify its effectiveness.

The previous MovingMNIST dataset [24, 23] only moves the digits with a constant speed and is not suitable for evaluating different models' ability in capturing more complicated motion patterns. We thus design the *MovingMNIST++* dataset by extending MovingMNIST to allow random rotations, scale changes, and illumination changes. Each frame is of size $64 \times 64$ and contains three moving digits. We use 10 frames as input to predict the next 10 frames. As the frames have illumination changes, we use MSE instead of cross-entropy for training and evaluation [2]. We train all models using the Adam optimizer [14] with learning rate equal to $10^{-4}$ and momentum equal to 0.5. For the RNN models, we use the encoding-forecasting structure introduced previously with three RNN layers. All RNNs are either ConvGRU or TrajGRU and all use the same set of hyperparameters. For TrajGRU, we initialize the weight of the output layer of the structure generating network to zero. The strides of the middle downsampling and upsampling layers are chosen to be 2. The numbers of filters for the three RNNs are $64, 96, 96$ respectively. For the DFN model, we replace the output layer of ConvGRU with a $11 \times 11$ local filter and transform the previous frame to get the prediction. For the RNN models, we train them for 200,000 iterations with norm clipping threshold equal to 1 and batch size equal to 4. For the CNN models, we train them for 100,000 iterations with norm clipping threshold equal to 50 and batch size equal to 32. The detailed experimental configuration of the models for the MovingMNIST++ experiment can be found in the appendix. We have also tried to use conditional GAN for the 2D and 3D models but have failed to get reasonable results.

Table 1 gives the results of different models on the same test set that contains 10,000 sequences. We train all models using three different seeds to report the standard deviation. We can find that TrajGRU with only 5 links outperforms ConvGRU with state-to-state kernel size $3 \times 3$ and dilation $2 \times 2$ (9 links). Also, the performance of TrajGRU improves as the number of links increases. TrajGRU with $L = 13$ outperforms ConvGRU with $7 \times 7$ state-to-state kernel and yet has fewer parameters. Another observation from the table is that DFN does not perform well in this synthetic dataset. This is because DFN uses softmax to enhance the sparsity of the learned local filters, which fails to model illumination change because the maximum value always gets smaller after convolving with a positive kernel whose weights sum up to 1. For DFN, when the pixel values get smaller, it is impossible for them to increase again. Figure 3 visualizes the learned structures of TrajGRU. We can see that the network has learned reasonable local link patterns.

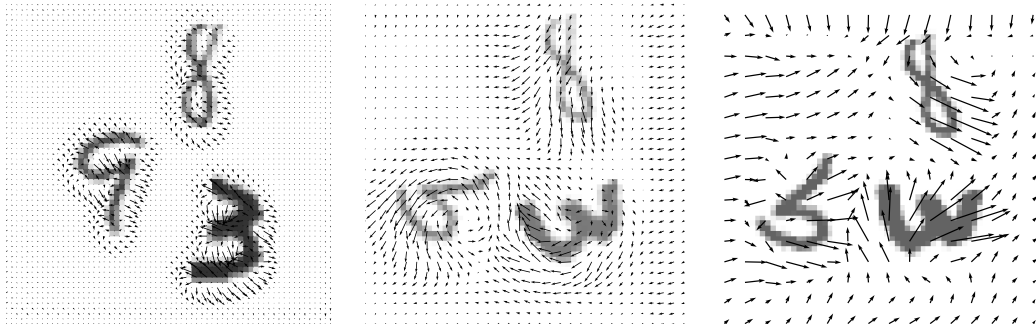

**Figure 3:** Selected links of TrajGRU-L13 at different frames and layers. We choose one of the 13 links and plot an arrow starting from each pixel to the pixel that is referenced by the link. From left to right we display the learned structure at the first, second and third layer of the encoder. The links displayed here have learned behaviour for rotations. We sub-sample the displayed links for the first layer for better readability. We include animations for all layers and links in the supplementary material. (Best viewed when zoomed in.)

## 5 Benchmark for Precipitation Nowcasting

### 5.1 HKO-7 Dataset

The HKO-7 dataset used in the benchmark contains radar echo data from 2009 to 2015 collected by HKO. The radar CAPPI reflectivity images, which have resolution of $480 \times 480$ pixels, are taken from an altitude of 2km and cover a $512\text{km} \times 512\text{km}$ area centered in Hong Kong. The data are recorded every 6 minutes and hence there are 240 frames per day. The raw logarithmic radar reflectivity factors are linearly transformed to pixel values via pixel $= \lfloor 255 \times \frac{\text{dBZ}+10}{70} + 0.5 \rfloor$ and are clipped to be between 0 and 255. The raw radar echo images generated by Doppler weather radar are noisy due to factors like ground clutter, sea clutter, anomalous propagation and electromagnetic interference [16]. To alleviate the impact of noise in training and evaluation, we filter the noisy pixels in the dataset and generate the noise masks by a two-stage process described in the appendix.

As rainfall events occur sparsely, we select the rainy days based on the rain barrel information to form our final dataset, which has 812 days for training, 50 days for validation and 131 days for testing. Our current treatment is close to the real-life scenario as we are able to train an additional model that classifies whether or not it will rain on the next day and applies our precipitation nowcasting model if this coarser-level model predicts that it will be rainy. The radar reflectivity values are converted to rainfall intensity values (mm/h) using the Z-R relationship: $\text{dBZ} = 10 \log a + 10b \log R$ where $R$ is the rain-rate level, $a = 58.53$ and $b = 1.56$. The overall statistics and the average monthly rainfall distribution of the HKO-7 dataset are given in the appendix.

### 5.2 Evaluation Methodology

As the radar echo maps arrive in a stream, nowcasting algorithms can apply online learning to adapt to the newly emerging spatiotemporal patterns. We propose two settings in our evaluation protocol: (1) the offline setting in which the algorithm always receives 5 frames as input and predicts 20 frames ahead, and (2) the online setting in which the algorithm receives segments of length 5 sequentially and predicts 20 frames ahead for each new segment received. The evaluation protocol is described more systematically in the appendix. The testing environment guarantees that the same set of sequences is tested in both the offline and online settings for fair comparison.

For both settings, we evaluate the skill scores for multiple thresholds that correspond to different rainfall levels to give an all-round evaluation of the algorithms' nowcasting performance. Table 2 shows the distribution of different rainfall levels in our dataset. We choose to use the thresholds 0.5, 2, 5, 10, 30 to calculate the CSI and Heidke Skill Score (HSS) [8]. For calculating the skill score at a specific threshold $\tau$, which is 0.5, 2, 5, 10 or 30, we first convert the pixel values in prediction and ground-truth to 0/1 by thresholding with $\tau$. We then calculate the TP (prediction=1, truth=1), FN (prediction=0, truth=1), FP (prediction=1, truth=0), and TN (prediction=0, truth=0). The CSI score is calculated as $\frac{\text{TP}}{\text{TP}+\text{FN}+\text{FP}}$ and the HSS score is calculated as $\frac{\text{TP}\times\text{TN}-\text{FN}\times\text{FP}}{(\text{TP}+\text{FN})(\text{FN}+\text{TN})+(\text{TP}+\text{FP})(\text{FP}+\text{TN})}$. During the computation, the masked points are ignored.

**Table 2:** Rain rate statistics in the HKO-7 benchmark.

| Rain Rate (mm/h) | | | Proportion (%) | Rainfall Level |
|---|---|---|---|---|
| $0 \leq$ | $x$ | $< 0.5$ | 90.25 | No / Hardly noticeable |
| $0.5 \leq$ | $x$ | $< 2$ | 4.38 | Light |
| $2 \leq$ | $x$ | $< 5$ | 2.46 | Light to moderate |
| $5 \leq$ | $x$ | $< 10$ | 1.35 | Moderate |
| $10 \leq$ | $x$ | $< 30$ | 1.14 | Moderate to heavy |
| $30 \leq$ | $x$ | | 0.42 | Rainstorm warning |

As shown in Table 2, the frequencies of different rainfall levels are highly imbalanced. We propose to use the weighted loss function to help solve this problem. Specifically, we assign a weight $w(x)$ to each pixel according to its rainfall intensity $x$: $w(x) = \begin{cases} 1, & x < 2 \\ 2, & 2 \leq x < 5 \\ 5, & 5 \leq x < 10 \\ 10, & 10 \leq x < 30 \\ 30, & x \geq 30 \end{cases}$ . Also, the masked pixels have weight 0. The resulting B-MSE and B-MAE scores are computed as B-MSE $= \frac{1}{N}\sum_{n=1}^{N}\sum_{i=1}^{480}\sum_{j=1}^{480} w_{n,i,j}(x_{n,i,j} - \hat{x}_{n,i,j})^2$ and B-MAE $= \frac{1}{N}\sum_{n=1}^{N}\sum_{i=1}^{480}\sum_{j=1}^{480} w_{n,i,j}|x_{n,i,j} - \hat{x}_{n,i,j}|$, where $N$ is the total number of frames and $w_{n,i,j}$ is the weight corresponding to the $(i,j)$th pixel in the $n$th frame. For the conventional MSE and MAE measures, we simply set all the weights to 1 except the masked points.

### 5.3 Evaluated Algorithms

We have evaluated seven nowcasting algorithms, including the simplest model which always predicts the last frame, two optical flow based methods (ROVER and its nonlinear variant), and four deep learning methods (TrajGRU, ConvGRU, 2D CNN, and 3D CNN). Specifically, we have evaluated the performance of deep learning models in the online setting by fine-tuning the algorithms using AdaGrad [4] with learning rate equal to $10^{-4}$. We optimize the sum of B-MSE and B-MAE during offline training and online fine-tuning. During the offline training process, all models are optimized by the Adam optimizer with learning rate equal to $10^{-4}$ and momentum equal to $0.5$ and we train these models with early-stopping on the sum of B-MSE and B-MAE. For RNN models, the training batch size is set to 4. For the CNN models, the training batch size is set to 8. For TrajGRU and ConvGRU models, we use a 3-layer encoding-forecasting structure with the number of filters for the RNNs set to $64, 192, 192$. We use kernel size equal to $5 \times 5, 5 \times 5, 3 \times 3$ for the ConvGRU models while the number of links is set to $13, 13, 9$ for the TrajGRU model. We also train the ConvGRU model with the original MSE and MAE loss, which is named "ConvGRU-nobal", to evaluate the improvement by training with the B-MSE and B-MAE loss. The other model configurations including ROVER, ROVER-nonlinear and deep models are included in the appendix.

### 5.4 Evaluation Results

The overall evaluation results are summarized in Table 3. In order to analyze the confidence interval of the results, we train 2D CNN, 3D CNN, ConvGRU and TrajGRU models using three different random seeds and report the standard deviation in Table 4. We find that training with balanced loss functions is essential for good nowcasting performance of heavier rainfall. The ConvGRU model that is trained without balanced loss, which best represents the model in [23], has worse nowcasting score than the optical flow based methods at the 10mm/h and 30mm/h thresholds. Also, we find that all the deep learning models that are trained with the balanced loss outperform the optical flow based models. Among the deep learning models, TrajGRU performs the best and 3D CNN outperforms 2D CNN, which shows that an appropriate network structure is crucial to achieving good performance. The improvement of TrajGRU over the other models is statistically significant because the differences in B-MSE and B-MAE are larger than three times their standard deviation. Moreover, the performance with online fine-tuning enabled is consistently better than that without online fine-tuning, which verifies the effectiveness of online learning at least for this task.

**Table 3:** HKO-7 benchmark result. We mark the best result within a specific setting with **bold face** and the second best result by underlining. Each cell contains the mean score of the 20 predicted frames. In the online setting, all algorithms have used the online learning strategy described in the paper. '↑' means that the score is higher the better while '↓' means that the score is lower the better. '$r \geq \tau$' means the skill score at the $\tau$mm/h rainfall threshold. For 2D CNN, 3D CNN, ConvGRU and TrajGRU models, we train the models with three different random seeds and report the mean scores.

| Algorithms | CSI ↑ | | | | | HSS ↑ | | | | | B-MSE ↓ | B-MAE ↓ |
|---|---|---|---|---|---|---|---|---|---|---|---|---|
| | $r \geq 0.5$ | $r \geq 2$ | $r \geq 5$ | $r \geq 10$ | $r \geq 30$ | $r \geq 0.5$ | $r \geq 2$ | $r \geq 5$ | $r \geq 10$ | $r \geq 30$ | | |
| | | | | | Offline Setting | | | | | | | |
| Last Frame | 0.4022 | 0.3266 | 0.2401 | 0.1574 | 0.0692 | 0.5207 | 0.4531 | 0.3582 | 0.2512 | 0.1193 | 15274 | 28042 |
| ROVER + Linear | 0.4762 | 0.4089 | 0.3151 | 0.2146 | 0.1067 | 0.6038 | 0.5473 | 0.4516 | 0.3301 | 0.1762 | 11651 | 23437 |
| ROVER + Non-linear | 0.4655 | 0.4074 | 0.3226 | 0.2164 | 0.0951 | 0.5896 | 0.5436 | 0.4590 | 0.3318 | 0.1576 | 10945 | 22857 |
| 2D CNN | 0.5095 | 0.4396 | 0.3406 | 0.2392 | 0.1093 | 0.6366 | 0.5809 | 0.4851 | 0.3690 | 0.1885 | 7332 | 18091 |
| 3D CNN | 0.5109 | 0.4411 | 0.3415 | 0.2424 | 0.1185 | 0.6334 | 0.5825 | 0.4862 | 0.3734 | 0.2034 | 7202 | 17593 |
| ConvGRU-nobal | 0.5476 | 0.4661 | 0.3526 | 0.2138 | 0.0712 | 0.6756 | 0.6094 | 0.4981 | 0.3286 | 0.1160 | 9087 | 19642 |
| ConvGRU | 0.5489 | 0.4731 | 0.3720 | 0.2789 | 0.1776 | 0.6701 | 0.6104 | 0.5163 | 0.4159 | 0.2893 | 5951 | 15000 |
| TrajGRU | **0.5528** | **0.4759** | **0.3751** | **0.2835** | **0.1856** | **0.6731** | **0.6126** | **0.5192** | **0.4207** | **0.2996** | **5816** | **14675** |
| | | | | | Online Setting | | | | | | | |
| 2D CNN | 0.5112 | 0.4363 | 0.3364 | 0.2435 | 0.1263 | 0.6365 | 0.5756 | 0.4790 | 0.3744 | 0.2162 | 6654 | 17071 |
| 3D CNN | 0.5106 | 0.4344 | 0.3345 | 0.2427 | 0.1299 | 0.6355 | 0.5736 | 0.4766 | 0.3733 | 0.2220 | 6690 | 16903 |
| ConvGRU | 0.5511 | 0.4737 | 0.3742 | 0.2843 | 0.1837 | 0.6712 | 0.6105 | 0.5183 | 0.4226 | 0.2981 | 5724 | 14772 |
| TrajGRU | **0.5563** | **0.4798** | **0.3808** | **0.2914** | **0.1933** | **0.6760** | **0.6164** | **0.5253** | **0.4308** | **0.3111** | **5589** | **14465** |

**Table 4:** Confidence intervals of selected deep models in the HKO-7 benchmark. We train 2D CNN, 3D CNN, ConvGRU and TrajGRU using three different random seeds and report the standard deviation of the test scores.

| Algorithms | CSI | | | | | HSS | | | | | B-MSE | B-MAE |
|---|---|---|---|---|---|---|---|---|---|---|---|---|
| | $r \geq 0.5$ | $r \geq 2$ | $r \geq 5$ | $r \geq 10$ | $r \geq 30$ | $r \geq 0.5$ | $r \geq 2$ | $r \geq 5$ | $r \geq 10$ | $r \geq 30$ | | |
| | | | | | Offline Setting | | | | | | | |
| 2D CNN | 0.0032 | 0.0023 | 0.0015 | 0.0001 | 0.0025 | 0.0032 | 0.0025 | 0.0018 | 0.0003 | 0.0043 | 90 | 95 |
| 3D CNN | 0.0043 | 0.0027 | 0.0016 | 0.0024 | 0.0024 | 0.0042 | 0.0028 | 0.0018 | 0.0031 | 0.0041 | 44 | 26 |
| ConvGRU | 0.0022 | 0.0018 | 0.0031 | 0.0008 | 0.0022 | 0.0022 | 0.0021 | 0.0040 | 0.0010 | 0.0038 | 52 | 81 |
| TrajGRU | 0.0020 | 0.0024 | 0.0025 | 0.0031 | 0.0031 | 0.0019 | 0.0024 | 0.0028 | 0.0039 | 0.0045 | 18 | 32 |
| | | | | | Online Setting | | | | | | | |
| 2D CNN | 0.0002 | 0.0005 | 0.0002 | 0.0002 | 0.0012 | 0.0002 | 0.0005 | 0.0002 | 0.0003 | 0.0019 | 12 | 12 |
| 3D CNN | 0.0004 | 0.0003 | 0.0002 | 0.0003 | 0.0008 | 0.0004 | 0.0004 | 0.0003 | 0.0004 | 0.0001 | 23 | 27 |
| ConvGRU | 0.0006 | 0.0012 | 0.0017 | 0.0019 | 0.0024 | 0.0006 | 0.0012 | 0.0019 | 0.0023 | 0.0031 | 30 | 69 |
| TrajGRU | 0.0008 | 0.0004 | 0.0002 | 0.0002 | 0.0002 | 0.0007 | 0.0004 | 0.0002 | 0.0002 | 0.0003 | 10 | 20 |

**Table 5:** Kendall's $\tau$ coefficients between skill scores. Higher absolute value indicates stronger correlation. The numbers with the largest absolute values are shown in **bold face**.

| Skill Scores | CSI | | | | | HSS | | | | |
|---|---|---|---|---|---|---|---|---|---|---|
| | $r \geq 0.5$ | $r \geq 2$ | $r \geq 5$ | $r \geq 10$ | $r \geq 30$ | $r \geq 0.5$ | $r \geq 2$ | $r \geq 5$ | $r \geq 10$ | $r \geq 30$ |
| MSE | -0.24 | -0.39 | -0.39 | -0.07 | -0.01 | -0.33 | -0.42 | -0.39 | -0.06 | 0.01 |
| MAE | -0.41 | -0.57 | -0.55 | -0.25 | -0.27 | -0.50 | **-0.60** | -0.55 | -0.24 | -0.26 |
| B-MSE | -0.70 | -0.57 | **-0.61** | **-0.86** | -0.84 | -0.62 | -0.55 | **-0.61** | **-0.86** | -0.84 |
| B-MAE | **-0.74** | **-0.59** | -0.58 | -0.82 | **-0.92** | **-0.67** | -0.57 | -0.59 | -0.83 | **-0.92** |

Based on the evaluation results, we also compute the Kendall's $\tau$ coefficients [13] between the MSE, MAE, B-MSE, B-MAE and the CSI, HSS at different thresholds. As shown in Table 5, B-MSE and B-MAE have stronger correlation with the CSI and HSS in most cases.

# 6 Conclusion and Future Work

In this paper, we have provided the first large-scale benchmark for precipitation nowcasting and have proposed a new TrajGRU model with the ability of learning the recurrent connection structure. We have shown TrajGRU is more efficient in capturing the spatiotemporal correlations than ConvGRU. For future work, we plan to test if TrajGRU helps improve other spatiotemporal learning tasks like visual object tracking and video segmentation. We will also try to build an operational nowcasting system using the proposed algorithm.

## Acknowledgments

This research has been supported by General Research Fund 16207316 from the Research Grants Council and Innovation and Technology Fund ITS/205/15FP from the Innovation and Technology Commission in Hong Kong. The first author has also been supported by the Hong Kong PhD Fellowship.

## Footnotes

[1]The radar echo maps are Constant Altitude Plan Position Indicator (CAPPI) images which can be converted to rainfall intensity maps using the Marshall-Palmer relationship or Z-R relationship [19].

[2]The MSE for the MovingMNIST++ experiment is averaged by both the frame size and the length of the predicted sequence.

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
