[Supplementary Material]

# Appendix of "Deep Learning for Precipitation Nowcasting: A Benchmark and A New Model"

## A    Weight Initialization

The weights and biases of all models are initialized with the MSRA initializer [1] except that the weights and biases of the structure generating network in TrajGRUs are initialized to be zero.

## B    Structure Generating Network in TrajGRU

The structure generating network takes the concatenation of the state tensor and the input tensor as the input. We fix the network to have two convolution layers. The first convolution layer uses $5 \times 5$ kernel size, $2 \times 2$ padding size, 32 filters and uses the leaky ReLU activation. The second convolution layer uses $5 \times 5$ kernel size, $2 \times 2$ padding and $2L$ filters where $L$ is the number of links.

## C    Details about the MovingMNIST++ Experiment

### C.1    Generation Process

For each sequence, we choose three digits randomly from the MNIST dataset[1]. Each digit will move, rotate, scale up or down at a randomly sampled speed. Also, we multiply the pixel values by an illumination factor every time to make the digits have time-varying appearances. The hyperparameters of the generation process are given in Table 1. In our experiment, we always generate a length-20 sequence and use the first 10 frames to predict the last 10 frames.

**Table 1:** Hyperparameters of the MovingMNIST++ dataset. We choose the velocity, scaling factor, rotation angle and illumination factor uniformly within the range listed in the table.

| Hyperparameter | Value |
|---|---|
| Number of digits | 3 |
| Frame size | $64 \times 64$ |
| Velocity | $[0, 3.6)$ |
| Scaling factor | $[\frac{1}{1.1}, 1.1)$ |
| Rotation angle | $[\frac{-\pi}{12}, \frac{\pi}{12})$ |
| Illumination factor | $[0.6, 1.0)$ |

### C.2    Network Structures

The general structure of the 2D CNN, 3D CNN and the DFN model used in the paper are illustrated in Figure 1. We always use batch normalization [2] in 2D and 3D CNNs.

The detailed network configurations of 2D CNN, 3D CNN, ConvGRU, DFN and TrajGRU are described in Table 2, 3, 4, 5, 6.

**(a)** Illustration of the 2D/3D CNNs used in the paper. In this example, we use 4 convolution layers to get the representation of the 5 input frames, which is further used to forecast the 5 future frames. We use either 2D convolution or 3D convolution in the encoder and the forecaster.

**(b)** Illustration of the DFN model used in the paper. In this example, we use 2 frames to predict 2 frames. The $\hat{S}$s are the predicted local filters, which are used to transform the last input frame or the previous predicted frame. We use ConvGRU as the RNN model in the experiment.

**Figure 1:** Illustration of the 2D CNN, 3D CNN and DFN models used in the paper.

**Table 2:** The details of the 2D CNN model. The two dimensions in kernel, stride, pad and other features represent for height and width. We set the base filter number $c$ to 70. We derive the 2D model from the 3D model by multiplying the number of channels with the respective kernel size of the 3D model. The 10 channels in the input of 'enc1' and the output of 'vid5' correspond to the input and output frames, respectively.

| Name | Kernel | Stride | Pad | Ch I/O | In Res | Out Res | Type | Input |
|------|--------|--------|-----|--------|--------|---------|------|-------|
| enc1 | $4 \times 4$ | $2 \times 2$ | $1 \times 1$ | $10/4c$ | $64 \times 64$ | $32 \times 32$ | Conv | in |
| enc2 | $4 \times 4$ | $2 \times 2$ | $1 \times 1$ | $4c/8c$ | $32 \times 32$ | $16 \times 16$ | Conv | enc1 |
| enc3 | $4 \times 4$ | $2 \times 2$ | $1 \times 1$ | $8c/12c$ | $16 \times 16$ | $8 \times 8$ | Conv | enc2 |
| enc4 | $4 \times 4$ | $2 \times 2$ | $1 \times 1$ | $12c/16c$ | $8 \times 8$ | $4 \times 4$ | Conv | enc3 |
| vid1 | $1 \times 1$ | $1 \times 1$ | $0 \times 0$ | $16c/16c$ | $4 \times 4$ | $4 \times 4$ | Deconv | enc4 |
| vid2 | $4 \times 4$ | $2 \times 2$ | $1 \times 1$ | $16c/16c$ | $4 \times 4$ | $8 \times 8$ | Deconv | vid1 |
| vid3 | $4 \times 4$ | $2 \times 2$ | $1 \times 1$ | $16c/8c$ | $8 \times 8$ | $16 \times 16$ | Deconv | vid2 |
| vid4 | $4 \times 4$ | $2 \times 2$ | $1 \times 1$ | $8c/4c$ | $16 \times 16$ | $32 \times 32$ | Deconv | vid3 |
| vid5 | $4 \times 4$ | $2 \times 2$ | $1 \times 1$ | $4c/10$ | $32 \times 32$ | $64 \times 64$ | Deconv | vid4 |

**Table 3:** The details of the 3D CNN model. The three dimensions in kernel, stride, pad and other features represent for depth, height and width. We set the base filter number $c$ to 128.

| Name | Kernel | Stride | Pad | Ch I/O | In Res | Out Res | Type | Input |
|------|--------|--------|-----|--------|--------|---------|------|-------|
| enc1 | $4 \times 4 \times 4$ | $2 \times 2 \times 2$ | $1 \times 1 \times 1$ | $1/c$ | $10 \times 64 \times 64$ | $5 \times 32 \times 32$ | Conv | in |
| enc2 | $4 \times 4 \times 4$ | $2 \times 2 \times 2$ | $1 \times 1 \times 1$ | $c/2c$ | $5 \times 32 \times 32$ | $2 \times 16 \times 16$ | Conv | enc1 |
| enc3 | $4 \times 4 \times 4$ | $2 \times 2 \times 2$ | $1 \times 1 \times 1$ | $2c/3c$ | $2 \times 16 \times 16$ | $1 \times 8 \times 8$ | Conv | enc2 |
| enc4 | $4 \times 4 \times 4$ | $2 \times 2 \times 2$ | $2 \times 1 \times 1$ | $3c/4c$ | $1 \times 8 \times 8$ | $1 \times 4 \times 4$ | Conv | enc3 |
| vid1 | $2 \times 1 \times 1$ | $1 \times 1 \times 1$ | $0 \times 0 \times 0$ | $4c/8c$ | $1 \times 4 \times 4$ | $2 \times 4 \times 4$ | Deconv | enc4 |
| vid2 | $4 \times 4 \times 4$ | $2 \times 2 \times 2$ | $1 \times 1 \times 1$ | $8c/4c$ | $2 \times 4 \times 4$ | $4 \times 8 \times 8$ | Deconv | vid1 |
| vid3 | $4 \times 4 \times 4$ | $2 \times 2 \times 2$ | $2 \times 1 \times 1$ | $4c/2c$ | $4 \times 8 \times 8$ | $6 \times 16 \times 16$ | Deconv | vid2 |
| vid4 | $4 \times 4 \times 4$ | $2 \times 2 \times 2$ | $2 \times 1 \times 1$ | $2c/c$ | $6 \times 16 \times 16$ | $10 \times 32 \times 32$ | Deconv | vid3 |
| vid5 | $3 \times 4 \times 4$ | $1 \times 2 \times 2$ | $1 \times 1 \times 1$ | $c/1$ | $10 \times 32 \times 32$ | $10 \times 64 \times 64$ | Deconv | vid4 |

**Table 4:** The details of the ConvGRU model. The 'In Kernel', 'In Stride' and 'In Pad' are the kernel, stride and padding in the input-to-state convolution. 'State Ker.' and 'State Dila.' are the kernel size and dilation size of the state-to-state convolution. We set $k$ and $d$ as stated in the paper. The 'In State' is the initial state of the RNN layer.

| Name | In Kernel | In Stride | In Pad | State Ker. | State Dila. | Ch I/O | In Res | Out Res | Type | In | In State |
|---|---|---|---|---|---|---|---|---|---|---|---|
| econv1 | $3 \times 3$ | $1 \times 1$ | $1 \times 1$ | - | - | 4/16 | $64 \times 64$ | $64 \times 64$ | Conv | in | - |
| ernn1 | $3 \times 3$ | $1 \times 1$ | $1 \times 1$ | $k \times k$ | $d \times d$ | 16/64 | $64 \times 64$ | $64 \times 64$ | ConvGRU | econv1 | - |
| edown1 | $3 \times 3$ | $2 \times 2$ | $1 \times 1$ | - | - | 64/64 | $64 \times 64$ | $32 \times 32$ | Conv | ernn1 | - |
| ernn2 | $3 \times 3$ | $1 \times 1$ | $1 \times 1$ | $k \times k$ | $d \times d$ | 64/96 | $32 \times 32$ | $32 \times 32$ | ConvGRU | edown1 | - |
| edown2 | $3 \times 3$ | $2 \times 2$ | $1 \times 1$ | - | - | 96/96 | $32 \times 32$ | $16 \times 16$ | Conv | ernn2 | - |
| ernn3 | $3 \times 3$ | $1 \times 1$ | $1 \times 1$ | $k \times k$ | $d \times d$ | 96/96 | $16 \times 16$ | $16 \times 16$ | ConvGRU | edown2 | - |
| frnn1 | $3 \times 3$ | $1 \times 1$ | $1 \times 1$ | $k \times k$ | $d \times d$ | 96/96 | $16 \times 16$ | $16 \times 16$ | ConvGRU | - | ernn3 |
| fup1 | $4 \times 4$ | $2 \times 2$ | $1 \times 1$ | - | - | 96/96 | $16 \times 16$ | $32 \times 32$ | Deconv | frnn1 | - |
| frnn2 | $3 \times 3$ | $1 \times 1$ | $1 \times 1$ | $k \times k$ | $d \times d$ | 96/96 | $32 \times 32$ | $32 \times 32$ | ConvGRU | fup1 | ernn2 |
| fup2 | $4 \times 4$ | $2 \times 2$ | $1 \times 1$ | - | - | 96/96 | $32 \times 32$ | $64 \times 64$ | Deconv | frnn2 | - |
| frnn3 | $3 \times 3$ | $1 \times 1$ | $1 \times 1$ | $k \times k$ | $d \times d$ | 96/64 | $64 \times 64$ | $64 \times 64$ | ConvGRU | fup2 | ernn1 |
| fconv4 | $3 \times 3$ | $1 \times 1$ | $1 \times 1$ | - | - | 64/16 | $64 \times 64$ | $64 \times 64$ | Conv | frnn3 | - |
| fconv5 | $1 \times 1$ | $1 \times 1$ | $0 \times 0$ | - | - | 16/1 | $64 \times 64$ | $64 \times 64$ | Conv | fconv4 | - |

**Table 5:** The details of the DFN model. The output of the 'fconv4' layer will be used to transform the previous prediction or the last input frame. All hyperparameters have the same meaning as in Table 4.

| Name | In Kernel | In Stride | In Pad | State Ker. | State Dila. | Ch I/O | In Res | Out Res | Type | In | In State |
|---|---|---|---|---|---|---|---|---|---|---|---|
| econv1 | $3 \times 3$ | $1 \times 1$ | $1 \times 1$ | - | - | 4/16 | $64 \times 64$ | $64 \times 64$ | Conv | in | - |
| ernn1 | $3 \times 3$ | $1 \times 1$ | $1 \times 1$ | $k \times k$ | $d \times d$ | 16/64 | $64 \times 64$ | $64 \times 64$ | ConvGRU | econv1 | - |
| edown1 | $3 \times 3$ | $2 \times 2$ | $1 \times 1$ | - | - | 64/64 | $64 \times 64$ | $32 \times 32$ | Conv | ernn1 | - |
| ernn2 | $3 \times 3$ | $1 \times 1$ | $1 \times 1$ | $k \times k$ | $d \times d$ | 64/96 | $32 \times 32$ | $32 \times 32$ | ConvGRU | edown1 | - |
| edown2 | $3 \times 3$ | $2 \times 2$ | $1 \times 1$ | - | - | 96/96 | $32 \times 32$ | $16 \times 16$ | Conv | ernn2 | - |
| ernn3 | $3 \times 3$ | $1 \times 1$ | $1 \times 1$ | $k \times k$ | $d \times d$ | 96/96 | $16 \times 16$ | $16 \times 16$ | ConvGRU | edown2 | - |
| frnn1 | $3 \times 3$ | $1 \times 1$ | $1 \times 1$ | $k \times k$ | $d \times d$ | 96/96 | $16 \times 16$ | $16 \times 16$ | ConvGRU | - | ernn3 |
| fup1 | $4 \times 4$ | $2 \times 2$ | $1 \times 1$ | - | - | 96/96 | $16 \times 16$ | $32 \times 32$ | Deconv | frnn1 | - |
| frnn2 | $3 \times 3$ | $1 \times 1$ | $1 \times 1$ | $k \times k$ | $d \times d$ | 96/96 | $32 \times 32$ | $32 \times 32$ | ConvGRU | fup1 | ernn2 |
| fup2 | $4 \times 4$ | $2 \times 2$ | $1 \times 1$ | - | - | 96/96 | $32 \times 32$ | $64 \times 64$ | Deconv | frnn2 | - |
| frnn3 | $3 \times 3$ | $1 \times 1$ | $1 \times 1$ | $k \times k$ | $d \times d$ | 96/64 | $64 \times 64$ | $64 \times 64$ | ConvGRU | fup2 | ernn1 |
| fconv4 | $3 \times 3$ | $1 \times 1$ | $1 \times 1$ | - | - | 64/121 | $64 \times 64$ | $64 \times 64$ | Conv | frnn3 | - |

**Table 6:** The details of the TrajGRU model. 'L' is the number of links in the state-to-state transition. We set $l$ as stated in the paper. All other hyperparameters have the same meaning as in Table 4.

| Name | In Kernel | In Stride | In Pad | L | Ch I/O | In Res | Out Res | Type | In | In State |
|---|---|---|---|---|---|---|---|---|---|---|---|
| econv1 | $3 \times 3$ | $1 \times 1$ | $1 \times 1$ | - | 4/16 | $64 \times 64$ | $64 \times 64$ | Conv | in | - |
| ernn1 | $3 \times 3$ | $1 \times 1$ | $1 \times 1$ | $l$ | 16/64 | $64 \times 64$ | $64 \times 64$ | TrajGRU | econv1 | - |
| edown1 | $3 \times 3$ | $2 \times 2$ | $1 \times 1$ | - | 64/64 | $64 \times 64$ | $32 \times 32$ | Conv | ernn1 | - |
| ernn2 | $3 \times 3$ | $1 \times 1$ | $1 \times 1$ | $l$ | 64/96 | $32 \times 32$ | $32 \times 32$ | TrajGRU | edown1 | - |
| edown2 | $3 \times 3$ | $2 \times 2$ | $1 \times 1$ | - | 96/96 | $32 \times 32$ | $16 \times 16$ | Conv | ernn2 | - |
| ernn3 | $3 \times 3$ | $1 \times 1$ | $1 \times 1$ | $l$ | 96/96 | $16 \times 16$ | $16 \times 16$ | TrajGRU | edown2 | - |
| frnn1 | $3 \times 3$ | $1 \times 1$ | $1 \times 1$ | $l$ | 96/96 | $16 \times 16$ | $16 \times 16$ | TrajGRU | - | ernn3 |
| fup1 | $4 \times 4$ | $2 \times 2$ | $1 \times 1$ | - | 96/96 | $16 \times 16$ | $32 \times 32$ | Deconv | frnn1 | - |
| frnn2 | $3 \times 3$ | $1 \times 1$ | $1 \times 1$ | $l$ | 96/96 | $32 \times 32$ | $32 \times 32$ | TrajGRU | fup1 | ernn2 |
| fup2 | $4 \times 4$ | $2 \times 2$ | $1 \times 1$ | - | 96/96 | $32 \times 32$ | $64 \times 64$ | Deconv | frnn2 | - |
| frnn3 | $3 \times 3$ | $1 \times 1$ | $1 \times 1$ | $l$ | 96/64 | $64 \times 64$ | $64 \times 64$ | TrajGRU | fup2 | ernn1 |
| fconv4 | $3 \times 3$ | $1 \times 1$ | $1 \times 1$ | - | 64/16 | $64 \times 64$ | $64 \times 64$ | Conv | frnn3 | - |
| fconv5 | $1 \times 1$ | $1 \times 1$ | $0 \times 0$ | - | 16/1 | $64 \times 64$ | $64 \times 64$ | Conv | fconv4 | - |

# D   Details about the HKO-7 Benchmark

## D.1   Overall Data Statistics

The overall statistics of the HKO-7 dataset is given in Figure 2 and Table 7.

**Figure 2:** Average rainfall intensity of different months in the HKO-7 dataset.

**Table 7:** Overall statistics of the HKO-7 dataset.

| | Train | Validate | Test |
|---|---|---|---|
| Years | 2009-2014 | 2009-2014 | 2015 |
| #Days | 812 | 50 | 131 |
| #Frames | 192,168 | 11,736 | 31,350 |

## D.2 Denoising Process

We first remove the ground clutter and sun spikes, which appear at a fixed position, by detecting the out-lier locations in the image. For each in-boundary location $i$ in the frame, we use the ratio of its pixel value equal to $1, 2, ..., 255$ as the feature $x_i \sim R^{255}$ and estimate these features' sample mean $\hat{\mu} = \frac{\sum_{i=1}^{N} x_i}{N}$ and covariance matrix $\hat{S} = \frac{\sum_{i=1}^{N}(x_i-\mu)(x_i-\mu)^T}{N-1}$. We then calculate the Mahalanobis distance $D_M(x) = \sqrt{(x-\hat{\mu})^T \hat{S}^\dagger (x-\hat{\mu})^2}$ of these features using the estimated mean and covariance. Locations that have the Mahalanobis distances higher than the mean distance plus three times the standard deviation are classified as outliers. After out-lier detection, the $480 \times 480$ locations in the image are divided into 177316 inliers, 2824 outliers and 50260 out-of-boundary points. The outlier detection process is illustrated in Figure 3. After out-lier detection, we further remove other types of noise, like sea clutter, by filtering out the pixels with value smaller than 71 and larger than 0. Two examples that compare the original radar echo sequence and the denoised sequence are included in the attached "denoising" folder.

(a) Mahalanobis distance of a random portion of 10000 in-lier locations and 157 out-lier locations. The threshold is chosen to be the mean distance plus three times the standard deviation. (Best viewed in color.)

(b) Outlier locations that are excluded in learning and evaluation. The purple points are the out-of-boundary locations and the red points are the outliers. (Best viewed in color.)

**Figure 3:** Illustration of the outlier detection process and the final outlier mask obtained in HKO-7 dataset.

## D.3 Evaluation Protocol

We illustrate our evaluation protocol in Algorithm 1. We can choose the evaluation type to be 'offline' or 'online'. In the online setting, the model is able to store the previously seen sequences in a buffer and fine-tune the parameters using the sampled training batches from the buffer. For algorithms that are tested in the online setting in the paper, we sample the last 25 consecutive frames in the buffer to update the model if these frames are available. The buffer will be made empty once a new episode flag is received, which indicates that the newly observed 5-frame segment is not consecutive to the previous frames.

## D.4 Details of Optical Flow based Algorithms

For the ROVER algorithm, we use the same hyperparameters as [3]. For the ROVER-nonlinear algorithm, we follow the implementation in [4]. We first non-linearly transform the input frames and then calculate the optical flow based on the transformed frames.

**Algorithm 1** Evaluation protocol in the HKO-7 benchmark

```
 1: procedure HKO7TEST(model, type)
 2:     env ← GETENV(type)
 3:     while not env.end() do
 4:         I_{1:J}, f_e ← env.next()                          ▷ f_e indicates whether it is a new episode
 5:         model.store(I_{1:J}, f_e)
 6:         if type = online then
 7:             model.update()
 8:         Î_{J+1:J+K} ← model.predict()
 9:         env.upload(Î_{J+1:J+K})
10:     env.save()
```

## D.5 Network Structures

We use the general structure for 2D and 3D CNNs illustrated in Figure 1a. The network configurations of the 2D CNN, 3D CNN, ConvGRU and TrajGRU models are described in Table 8, 9, 10, 11.

**Table 8:** The details of the 2D CNN model. The two dimensions in kernel, stride, pad and other features represent for height and width. We set the base filter number $c$ to 70. We derive the 2D model from the 3D model by multiplying the number of channels with the respective kernel size of the 3D model. The first 5 and last 20 channels respectively correspond to the in- and output frames.

| Name | Kernel | Stride | Pad | Ch I/O | In Res | Out Res | Type | Input |
|------|--------|--------|-----|--------|--------|---------|------|-------|
| enc0 | $7 \times 7$ | $5 \times 5$ | $1 \times 1$ | $5/c$ | $480 \times 480$ | $96 \times 96$ | Conv | in |
| enc1 | $4 \times 4$ | $3 \times 3$ | $1 \times 1$ | $c/c$ | $96 \times 96$ | $32 \times 32$ | Conv | enc0 |
| enc2 | $4 \times 4$ | $2 \times 2$ | $1 \times 1$ | $c/8c$ | $32 \times 32$ | $16 \times 16$ | Conv | enc1 |
| enc3 | $4 \times 4$ | $2 \times 2$ | $1 \times 1$ | $8c/12c$ | $16 \times 16$ | $8 \times 8$ | Conv | enc2 |
| enc4 | $4 \times 4$ | $2 \times 2$ | $1 \times 1$ | $12c/16c$ | $8 \times 8$ | $4 \times 4$ | Conv | enc3 |
| vid1 | $1 \times 1$ | $1 \times 1$ | $0 \times 0.$ | $16c/16c$ | $4 \times 4$ | $4 \times 4$ | Deconv | enc4 |
| vid2 | $4 \times 4$ | $2 \times 2$ | $1 \times 1.$ | $16c/16c$ | $4 \times 4$ | $8 \times 8$ | Deconv | vid1 |
| vid3 | $4 \times 4$ | $2 \times 2$ | $1 \times 1.$ | $16c/8c$ | $8 \times 8$ | $16 \times 16$ | Deconv | vid2 |
| vid4 | $4 \times 4$ | $2 \times 2$ | $1 \times 1.$ | $8c/4c$ | $16 \times 16$ | $32 \times 32$ | Deconv | vid3 |
| vid5 | $5 \times 5$ | $3 \times 3$ | $1 \times 1.$ | $4c/24$ | $32 \times 32$ | $96 \times 96$ | Deconv | vid4 |
| vid6 | $7 \times 7$ | $5 \times 5$ | $1 \times 1.$ | $24/24$ | $96 \times 96$ | $480 \times 480$ | Deconv | vid5 |
| vid7 | $3 \times 3$ | $1 \times 1$ | $1 \times 1.$ | $24/20$ | $480 \times 480$ | $480 \times 480$ | Deconv | vid6 |

**Table 9:** The details of the 3D CNN model. The three dimensions in kernel, stride, pad and other features represent for channel, height and width. We set the base filter number c to 128.

| Name | Kernel | Stride | Pad | Ch I/O | In Res | Out Res | Type | Input |
|------|--------|--------|-----|--------|--------|---------|------|-------|
| enc0 | $1 \times 7 \times 7$ | $1 \times 5 \times 5$ | $0 \times 1 \times 1$ | $1/c$ | $5 \times 480 \times 480$ | $5 \times 96 \times 96$ | Conv | in |
| enc1 | $1 \times 4 \times 4$ | $1 \times 3 \times 3$ | $0 \times 1 \times 1$ | $c/c$ | $5 \times 96 \times 96$ | $5 \times 32 \times 32$ | Conv | enc0 |
| enc2 | $4 \times 4 \times 4$ | $2 \times 2 \times 2$ | $1 \times 1 \times 1$ | $c/2c$ | $5 \times 32 \times 32$ | $2 \times 16 \times 16$ | Conv | enc1 |
| enc3 | $4 \times 4 \times 4$ | $2 \times 2 \times 2$ | $1 \times 1 \times 1$ | $2c/3c$ | $2 \times 16 \times 16$ | $1 \times 8 \times 8$ | Conv | enc2 |
| enc4 | $4 \times 4 \times 4$ | $2 \times 2 \times 2$ | $2 \times 1 \times 1$ | $3c/4c$ | $1 \times 8 \times 8$ | $1 \times 4 \times 4$ | Conv | enc3 |
| vid1 | $2 \times 1 \times 1$ | $1 \times 1 \times 1$ | $0 \times 0 \times 0.$ | $4c/8c$ | $1 \times 4 \times 4$ | $2 \times 4 \times 4$ | Deconv | enc4 |
| vid2 | $4 \times 4 \times 4$ | $2 \times 2 \times 2$ | $1 \times 1 \times 1.$ | $8c/4c$ | $2 \times 4 \times 4$ | $4 \times 8 \times 8$ | Deconv | vid1 |
| vid3 | $4 \times 4 \times 4$ | $2 \times 2 \times 2$ | $0 \times 1 \times 1.$ | $4c/2c$ | $4 \times 8 \times 8$ | $10 \times 16 \times 16$ | Deconv | vid2 |
| vid4 | $4 \times 4 \times 4$ | $2 \times 2 \times 2$ | $1 \times 1 \times 1.$ | $2c/c$ | $10 \times 16 \times 16$ | $20 \times 32 \times 32$ | Deconv | vid3 |
| vid5 | $3 \times 5 \times 5$ | $1 \times 3 \times 3$ | $1 \times 1 \times 1.$ | $c/8$ | $20 \times 32 \times 32$ | $20 \times 96 \times 96$ | Deconv | vid4 |
| vid6 | $3 \times 7 \times 7$ | $1 \times 5 \times 5$ | $1 \times 1 \times 1.$ | $8/8$ | $20 \times 96 \times 96$ | $20 \times 480 \times 480$ | Deconv | vid5 |
| vid7 | $3 \times 3 \times 3$ | $1 \times 1 \times 1$ | $1 \times 1 \times 1.$ | $8/1$ | $20 \times 480 \times 480$ | $20 \times 480 \times 480$ | Deconv | vid6 |

**Table 10:** The details of the ConvGRU model. All hyperparameters have the same meaning as in Table 4.

| Name | In Kernel | In Stride | In Pad | State Ker. | State Dila. | Ch I/O | In Res | Out Res | Type | In | In State |
|---|---|---|---|---|---|---|---|---|---|---|---|
| econv1 | $7 \times 7$ | $5 \times 5$ | $1 \times 1$ | - | - | 4/8 | $480 \times 480$ | $96 \times 96$ | Conv | in | - |
| ernn1 | $3 \times 3$ | $1 \times 1$ | $1 \times 1$ | $5 \times 5$ | $1 \times 1$ | 8/64 | $96 \times 96$ | $96 \times 96$ | ConvGRU | econv1 | - |
| edown1 | $5 \times 5$ | $3 \times 3$ | $1 \times 1$ | - | - | 64/64 | $96 \times 96$ | $32 \times 32$ | Conv | ernn1 | - |
| ernn2 | $3 \times 3$ | $1 \times 1$ | $1 \times 1$ | $5 \times 5$ | $1 \times 1$ | 64/192 | $32 \times 32$ | $32 \times 32$ | ConvGRU | edown1 | - |
| edown2 | $3 \times 3$ | $2 \times 2$ | $1 \times 1$ | - | - | 192/192 | $32 \times 32$ | $16 \times 16$ | Conv | ernn2 | - |
| ernn3 | $3 \times 3$ | $1 \times 1$ | $1 \times 1$ | $3 \times 3$ | $1 \times 1$ | 192/192 | $16 \times 16$ | $16 \times 16$ | ConvGRU | edown2 | - |
| frnn1 | $3 \times 3$ | $1 \times 1$ | $1 \times 1$ | $3 \times 3$ | $1 \times 1$ | 192/192 | $16 \times 16$ | $16 \times 16$ | ConvGRU | - | ernn3 |
| fup1 | $4 \times 4$ | $2 \times 2$ | $1 \times 1$ | - | - | 192/192 | $16 \times 16$ | $32 \times 32$ | Deconv | frnn1 | - |
| frnn2 | $3 \times 3$ | $1 \times 1$ | $1 \times 1$ | $5 \times 5$ | $1 \times 1$ | 192/192 | $32 \times 32$ | $32 \times 32$ | ConvGRU | fup1 | ernn2 |
| fup2 | $5 \times 5$ | $3 \times 3$ | $1 \times 1$ | - | - | 192/192 | $32 \times 32$ | $96 \times 96$ | Deconv | frnn2 | - |
| frnn3 | $3 \times 3$ | $1 \times 1$ | $1 \times 1$ | $5 \times 5$ | $1 \times 1$ | 192/64 | $96 \times 96$ | $96 \times 96$ | ConvGRU | fup2 | ernn1 |
| fdeconv4 | $7 \times 7$ | $5 \times 5$ | $1 \times 1$ | - | - | 64/8 | $96 \times 96$ | $480 \times 480$ | Deconv | frnn3 | - |
| fconv5 | $1 \times 1$ | $1 \times 1$ | $0 \times 0$ | - | - | 8/1 | $480 \times 480$ | $480 \times 480$ | Conv | fdeconv4 | - |

**Table 11:** The details of the TrajGRU model. All hyperparameters have the same meaning as in Table 6.

| Name | In Kernel | In Stride | In Pad | L | Ch I/O | In Res | Out Res | Type | In | In State |
|---|---|---|---|---|---|---|---|---|---|---|
| econv1 | $7 \times 7$ | $5 \times 5$ | $1 \times 1$ | - | 4/8 | $480 \times 480$ | $96 \times 96$ | Conv | in | - |
| ernn1 | $3 \times 3$ | $1 \times 1$ | $1 \times 1$ | 13 | 8/64 | $96 \times 96$ | $96 \times 96$ | TrajGRU | econv1 | - |
| edown1 | $5 \times 5$ | $3 \times 3$ | $1 \times 1$ | - | 64/64 | $96 \times 96$ | $32 \times 32$ | Conv | ernn1 | - |
| ernn2 | $3 \times 3$ | $1 \times 1$ | $1 \times 1$ | 13 | 64/192 | $32 \times 32$ | $32 \times 32$ | TrajGRU | edown1 | - |
| edown2 | $3 \times 3$ | $2 \times 2$ | $1 \times 1$ | - | 192/192 | $32 \times 32$ | $16 \times 16$ | Conv | ernn2 | - |
| ernn3 | $3 \times 3$ | $1 \times 1$ | $1 \times 1$ | 9 | 192/192 | $16 \times 16$ | $16 \times 16$ | TrajGRU | edown2 | - |
| frnn1 | $3 \times 3$ | $1 \times 1$ | $1 \times 1$ | 9 | 192/192 | $16 \times 16$ | $16 \times 16$ | TrajGRU | - | ernn3 |
| fup1 | $4 \times 4$ | $2 \times 2$ | $1 \times 1$ | - | 192/192 | $16 \times 16$ | $32 \times 32$ | Deconv | frnn1 | - |
| frnn2 | $3 \times 3$ | $1 \times 1$ | $1 \times 1$ | 13 | 192/192 | $32 \times 32$ | $32 \times 32$ | TrajGRU | fup1 | ernn2 |
| fup2 | $5 \times 5$ | $3 \times 3$ | $1 \times 1$ | - | 192/192 | $32 \times 32$ | $96 \times 96$ | Deconv | frnn2 | - |
| frnn3 | $3 \times 3$ | $1 \times 1$ | $1 \times 1$ | 13 | 192/64 | $96 \times 96$ | $96 \times 96$ | TrajGRU | fup2 | ernn1 |
| fdeconv4 | $7 \times 7$ | $5 \times 5$ | $1 \times 1$ | - | 64/8 | $96 \times 96$ | $480 \times 480$ | Deconv | frnn3 | - |
| fconv5 | $1 \times 1$ | $1 \times 1$ | $0 \times 0$ | - | 8/1 | $480 \times 480$ | $480 \times 480$ | Conv | fdeconv4 | - |

## Footnotes

[1]MNIST dataset:`http://yann.lecun.com/exdb/mnist/`

[2]We use Moore-Penrose pseudoinverse in the implementation.