[Reviews · NeurIPS 2017]

Reviewer 1



This paper introduces TrajGRU, an extension of the convolutional LSTM/GRU. Contrary to convLSTM/GRU, TrajGRU aims at learning location dependant filter support for each hidden state location. TrajGRU generates a flow field from the current input and previous hidden state and then warp the previous hidden states through bilinear sampling following this flow field. Author evaluate their proposal on a video generation on two datasets, MovingMNIST having 3 digits at the same time and HKO-7 nowcasting dataset, where TrajRU outperforms their convolutional counterpart. Few specific question/remarks: Did you compare TrajGRU with ConvGRU having a larger support than just a 5x5 kernels? Does the TrajGRU requires more computation than a convGRU due to its warping operation? It would be informative to provide the overall parameter, number of operation and running time for the models evaluated in the experiment section. Why did you trained the model for a fix number of epoch rather than doing early stopping, could the performances of some model be improved by stopping the training earlier? - Quality The paper seems technically sound. - Clarity The paper is clear overall. It would be nice to specify the warp method to have more complete view of the TrajGRU. Also it is not clear what are the number of examples and training/validation/test splits for the HKO-7 datasets. -Originality Few other works have explored the use of warping for video model. See "Spatio-temporal video autoencoder with differentiable memory" for instance. It would to compare/contrast TrajGRU to this approach. -Significance/Conclusion Designing model that learn good video representation is still an ongoing research problem. This paper propose a novel model that propose to learn the filter support in addition to filter weight which is an interesting step toward better video model. However, the approach is only tested so far on one synthetic dataset (MovingMnist) and one specialized nowcasting dataset. It would be good to see if this model lead to better video representation for more traditional video task such as human action classification with generic videos.

Reviewer 2



Summary This paper describes a new GRU-based architecture for precipitation nowcasting, a task which can be seen as a video prediction problem with a fixed camera position. The authors also describe a new benchmarking package for precipitation nowcasting and evaluate their model on both this benchmark and an altered version of moving MNIST. Technical quality I think this is a good application paper while the proposed architecture is interesting as well. For the sake of completeness I would have liked to see some comparisons with LSTM versions of the networks as well and with fully connected RNNs but the authors already did look at a nice selection of baselines. Clarity The paper is generally well-written and coherent in structure. While I found it easy to follow the definition of the ConvGRU, I found the description of the proposed TrajGRU much harder to understand. The most important change in the new model is the replacement of the state-to-state convolutions/multiplications with a transformation that seems to be a specific version of the type of module described by Jaderberg et al. (2015) (reference 10 in the paper). It is not clear to me how the structure generating network gamma produces the flow fields U and V exactly though. It would help if this was made more explicit and if there would be a more clear emphasis on the differences with the transformation module from Jaderberg et al. (2015). At least the goal of the trajectory selection mechanism itself is clear enough for the rest of the narrative to make sense. Novelty While the general architecture is the same as the ConvLSTM and the paper is somewhat application oriented, the idea of learning trajectories with spatial transformer inspired modules is quite original to me. Significance The new data set seems to be a useful benchmark but I know too little about precipitation nowcasting to judge this. I also don't know enough about the state-of-the-art in that field to judge the performance gains very well and since the benchmark is a new one I suppose time will need to tell. The work would have been stronger if there was also a comparison on another existing precipitation nowcasting benchmark. I think that the idea to learn sparsely connected temporal trajectories may have more widespread applications in other domains where the distributions to be modelled are of high dimensionality but display certain independence properties and redundance. pros: Interesting model Nice baselines and practical evaluation The paper is generally well-written. cons: The description and motivation of the model is a bit short and hard to follow. The novelty is somewhat limited compared to ConvGRU/LSTM EDIT: The rebuttal addressed some of the concerns I had about the paper and I think that the additional experiments also add to the overall quality so I increased my score.

Reviewer 3



This paper makes novel contributions in the following two aspects: proposing a new dataset for nowcasting (if the dataset goes public, the reviewer is not able to find such a claim in the paper though) and applying the idea from 'Spatial Transformer Network' in a convRNN setup to predict the next few frames given a recent history. The introduction is well written, and the problem is well motivated. The rest of paper is clearly structured, but with following two points that are less satisfactory: 1. The use of \mathcal{U}_t, and \mathcal{V}_t and their corresponding \gamma function deserves much more detailed discussion as, according to the rest of the paper, this is the only technical difference from the Spatial Transformer Network where a 2*3 affine transformation matrix is learned instead. It is quite unclear why such scheme is abandoned and resorted to continuous optical flow. 2. The experiments are to some certain degree convincing. The measured difference are often statistically not significant (my guess as a reviewer as no confidence interval is given). In addition, as baselines, Conv3Ds often underperform Conv2Ds, which the reviewer finds strange given the nature of the problem. The reviewer would like the authors to properly address my above concerns to revise the feedback. Addition: The author feedback helps to resolve my questions.